

# Wind Turbine Wake Measurements with Automatically Adjusting Scanning Trajectories in a Multi-Doppler Lidar Setup

Norman Wildmann[1], Nikola Vasiljevic[2], and Thomas Gerz[1]

[1]Deutsches Zentrum für Luft- und Raumfahrt e.V., Münchner Str. 20, Oberpfaffenhofen, Germany
[2]DTU Wind Energy, Technical University of Denmark, Frederiksborgvej 399, Building 118-VEA, 4000 Roskilde, Denmark
*Correspondence to:* Norman Wildmann (norman.wildmann@dlr.de)

**Abstract.** In the context of the Perdigão 2017 experiment, the German Aerospace Center (DLR) deployed three long-range scanning Doppler lidars with the dedicated purpose of investigating the wake of a single wind turbine at the experimental site. A novel method was established to investigate wake properties with ground-based lidars over a wide range of wind directions. For this method, the three lidars, which were space- and time-synchronized using the WindScanner software, were programmed to measure with crossing beams at individual points up to ten rotor diameters downstream the wind turbine. Every half hour, the measurement points were adapted to the current wind direction to obtain a high availability of wake measurements in changing wind conditions. The linearly independent radial velocities where the lidar beams intersect allow the calculation of the wind vector at those points. Two approaches to estimate the prevailing wind direction were tested throughout the campaign. In the first approach, VAD scans of one of the lidars were used to calculate a five-minute average of wind speed and wind direction every half hour, whereas later in the experiment, five-minute averages of sonic anemometer measurements of a meteorological mast close to the wind turbine became available in real-time and were used for the scanning adjustment. Results of wind speed deficit measurements are presented for two measurement days with varying westerly winds and it is evaluated how well the lidar beam intersection points match the actual wake location. The new method allowed to obtain wake measurements over the whole measurement period, whereas a static scanning setup would only have captured short periods of wake occurrences. The analysed cases reveal that state-of-the-art engineering models for wakes underestimate the actual wind speed deficit.

## 1   Introduction

At the present state-of-the-art of wind-energy research, the investigation of the flow field downstream a wind-energy converter (WEC), i.e. in its wake, is a topic of high relevance for the siting and operation of WECs, especially in a typical configuration of a wind farm with multiple collocated turbines. For this purpose it is important to understand how high static and dynamic loads of neighbouring turbines will be. This question cannot be universally answered without taking the atmospheric conditions into consideration, which have an effect on the wind speed deficit in the wake, its width and propagation path. In order to describe wake dynamics beyond the typical engineering models (Göçmen et al., 2016), high-resolution numerical models (e.g. large-eddy simulations, LES) have proven to be an adequate tool and can realistically reproduce wind turbine wakes in well-defined conditions (Englberger and Dörnbrack, 2016; Wu and Porté-Agel, 2011, 2012). Nonetheless, models need vali-



dation by real-world experiments. Capturing all the previously mentioned properties of a highly dynamic, yet small-scale and local atmospheric feature like a wind turbine wake is a highly challenging task for wind measurement technology. It requires instruments that can sample wind speeds in a volume of the atmosphere in a short time and in a flexible way. While different measurement systems have been used, such as radars (Hirth et al., 2012), sodars (Barthelmie et al., 2003) or small remotely

piloted aircraft (RPA, Wildmann et al., 2014), it is lidar technology of different categories (pulsed and continuous-wave) that has proven to be the most versatile tool to perform wake measurements because of the high availability and reliability, good resolution and flexible possibilities to probe the atmosphere especially with scanning systems.

Coherent Doppler lidar instruments for wind speed measurement have been widely used in atmospheric sciences since the late 20[th] century (Frehlich et al., 1998; Reitebuch et al., 2001; Smalikho, 2003). Because of their good performance within the

atmospheric boundary layer (ABL), their high availability and capability to measure continuously, they have become increasingly interesting for wind-energy research as well (Emeis et al., 2007; Frehlich and Kelley, 2008). Conically scanning lidars (operating in the so-called velocity azimuth display mode, short: VAD) are useful to measure vertical profiles of wind speed and wind direction in the range between 50 m and the boundary-layer top. They can also be used to estimate dissipation rate and turbulent kinetic energy (TKE, Kumer et al., 2016). More advanced scanning scenarios can be performed with instruments

that have full hemispherical scanning capabilities and thus allow to scan arbitrary volumes of the atmosphere. Such systems have been deployed in the past to measure wind turbine wakes by scanning through them either horizontally (in so-called plan-position Indicator mode, short: PPI), or vertically (in so-called range-height indicator mode, short: RHI) (Käsler et al., 2010; Smalikho et al., 2013) and thus showing their propagation path. A drawback of measurements with single lidars is that only radial wind velocities along the line-of-sight of the laser beam can be retrieved. To overcome this limitation, multiple

lidars can be deployed with intersecting beams that allow a geometric reconstruction of the meteorological wind components (so-called dual-Doppler technique for the application of two lidars or multi-Doppler as the more general term Calhoun et al., 2006; Choukulkar et al., 2016; Drechsel et al., 2009; Mann et al., 2009; Newman et al., 2016). Examples for multi-lidar measurements of wind turbine wakes are for example Iungo et al. (2013), who placed dual-Doppler measurement points in the wake to retrieve vertical and horizontal wind speed at these points. van Dooren et al. (2016) combined PPI scans to retrieve the

horizontal wind speeds at hub height in the wake, and its horizontal propagation.

Placing measurement points into the wake with ground-based lidars is not trivial, because wind direction and thus turbine yaw angle and wake position change over time and the time that a fixed measurement point in space is measuring the wake can thus be very limited. A good way to overcome this problem is to place the lidar on the nacelle as it was done by Bingöl et al. (2009); Trujillo et al. (2011); Aitken and Lundquist (2014). The logistics for a nacelle-based installation are however significantly

larger and a single lidar on the nacelle will not be able to retrieve three-dimensional wind vector measurements directly.

In this study, we propose a new measurement strategy for the ground-based lidars that aims at solving the issue of limited measurement time of the wake due the turbine yawing during the measurements. The measurement strategy employs three ground-based scanning lidars which scanning trajectories are synchronized and adapted in accordance to the prevailing wind direction to continuously measure a wind turbine wake. By being able to adapt to the prevailing wind direction the availability

of the wake measurements are expected to increase significantly in comparison to the static case.





The paper is structured as follows. Section 2 introduces the measurement campaign and the instrumental set-up, Sect. 3 describes the methods to deduce turbine wake characteristics. In Sect. 4 the results are presented and discussed whereas Sect. 5 concludes our study with a list of lessons learned and next steps for future work.

## 2 Experiment description

### 2.1 The Perdigão 2017 experiment

The Perdigão 2017 experiment (Fernando et al., 2018) is part of a series of campaigns associated to the NEWA (New European Wind Atlas) project (Mann et al., 2017) and took place in central Portugal in late spring and early summer 2017. The main goal of the experiment is to understand the flow field over two mountain ridges, which are nearly parallel to each other, with a distance of 1.4 km and a height of more than 200 m over the surrounding valley (see Fig. 1). The intensive operation period (IOP) was set from 01 May to 15 June. An extended measurement period with a reduced amount of equipment lasted for a whole year, i.e. from December 2016 until December 2017. On the South-West ridge of the Perdigão mountains, an Enercon E-82 WEC is installed. The turbine is designed for low wind speeds and has a cut-in speed of 2 m s$^{-1}$. Nominal wind speed is 12 m s$^{-1}$ with a rated power of 2 MW. The hub height is 78 m above ground level, the rotor diameter is 82 m. Up to now, data from the turbine are not available. At the same location, a first, smaller campaign was carried out in 2015 and is described in Vasiljević et al. (2017), while a detailed analysis of wind turbine wake measurements of this experiment is presented in Menke et al. (2018). Many of the lessons that were learned in that first campaign were considered in the design of the presented campaign.

### 2.2 DLR instrumentation and lidar scanning strategy

The German Aerospace Center (DLR) contributed to the experiment with three long-range Doppler lidar systems of type Leosphere Windcube 200S, upgraded to work in the WindScanner mode (Vasiljevic et al., 2016), a microwave radiometer of type HATPRO-5G and several microphones. The research goals with these instruments are to study weather dependent sound propagation of wind turbine noise, but also the investigation of wakes in different atmospheric stability regimes. In this study, wake measurements of the lidar instruments are presented. A summary of the key features of the lidar systems is given in Tab. 1.

Figure 1 shows the location of the three lidars, denoted #1, #2 and #3. The purpose for these locations is to have two instruments in line with the WEC in main wind direction, which is from South-West, one in the valley between the ridges and one on the distant mountain ridge, in order to be able to perform coplanar RHI scans. The third lidar could be used to cut the wake with RHI scans at several distances behind the turbine. The analysis of coplanar and wake cut scans is not part of this study. Good wake measurements with coplanar vertical scans are only possible in this setup if the wind is in a narrow angular range perpendicular to the ridges. For example, at an offset to the plane of 10°, the wake center will be 40 m from the scanning plane at a distance of three rotor diameters downstream the WEC and thus not detectable by the coplanar scans any more.



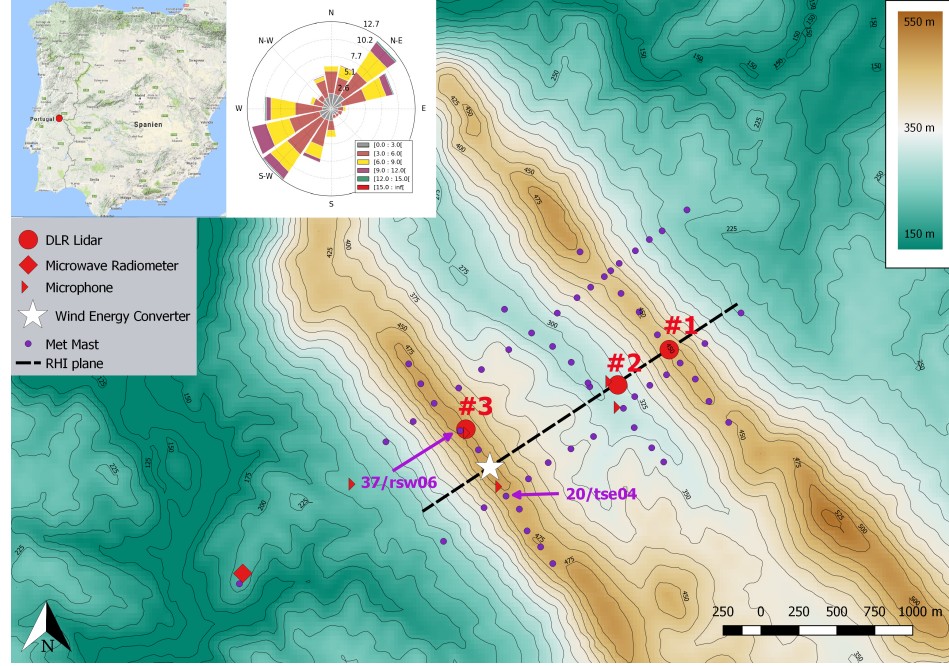

**Figure 1.** Map of the Perdigão site, including DLR instrumentation and meteorological mast locations. It also shows a wind rose for the sonic anemometer at 80 m on tower 20/tse04 for the IOP period. The dashed black line indicates the typical RHI scanning plane.

| Type | Pulsed Coherent Doppler Wind Lidar | | |
|---|---|---|---|
| Wave length | $1.54~\mu m$ | | |
| Measurement range | $-30 - 30$ m s$^{-1}$ | | |
| Dimensions | 995 x 810 x 1410 mm | | |
| Weight | 250 kg | | |
| Pulse | 100 ns | 200 ns | 400 ns |
| PRF | 40 kHz | 20 kHz | 10 kHz |
| FFT size | 64 | 128 | 256 |
| Physical resolution | 25 m | 50 m | 100 m |
| Typical range | < 2 km | < 5 km | < 10 km |

**Table 1.** Leosphere Windcube 200S, specification overview

The locations of the lidar systems however also allow to measure the wake in a wide area of Westerly wind directions, if the scanning trajectories are adapted accordingly. In this study, concepts are shown how the availability of wake measurements can be increased by automatic adaptation of scanning scenarios with respect to the prevailing wind direction. The goal is to determine the wind conditions up to ten rotor diameters downstream of the WEC at hub height. To achieve this, measurement

5 points are defined in space where the three lidar beams will cross in order to reconstruct the wind vector. These points are





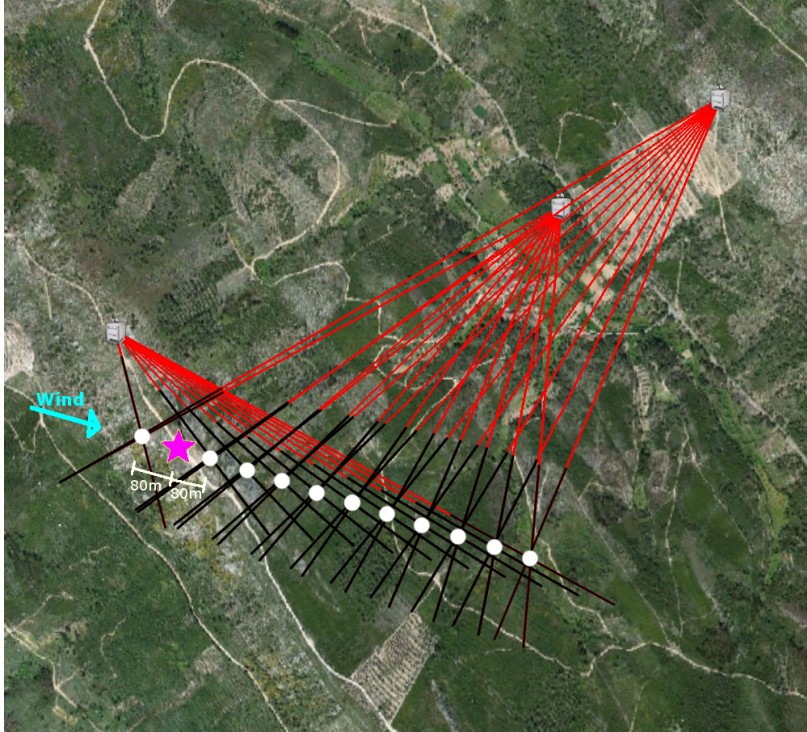

**Figure 2.** Visualization of multi-Doppler wake measurement scenario. Red are the lidar beams, black is the area with 2 m separated range gates, the pink star is the WEC and the white dots the center range gates where lidar beams cross.

placed equidistantly behind the wind turbine with a separation of one rotor diameter (i.e. 80 m). Additionally, on the later experiment days, one point was added one rotor diameter upstream, to measure the inflow of the WEC. In order to capture more of the wake with a single line-of-sight lidar measurement than just a single point, multiple range gates are added before and after the multi-Doppler measurement point. Fig. 2 depicts this scanning scenario.

## 2.3 Planning and control of complex trajectories

The Danish Technical University (DTU) developed a special software called WindScanner Client Software (WCS) for the Windcube scanning lidar systems for synchronizing multiple instruments in time and space allowing scenarios with multiple crossing points in space. This feature is not available with standard lidar systems. With WindScanner a synchronization to millisecond accuracy by GPS timing and a few meters spatial offset between the lidar beams by thorough calibration of the pointing direction and lidar position is feasible. The technical details of the software and hardware modifications to the Windcube lidar instrument is described in detail in Vasiljevic (2014); Vasiljevic et al. (2016).

While the WCS enables the low-level control of the lidars to synchronize predefined scenarios on multiple systems, it is up to the user to define these scenarios with matching azimuth, elevation and range gate distance for crossing lidar beams at




predefined points. For this purpose, the authors have developed a planning tool based on Java and the NASA World Wind virtual globe software development kit (NASA, 2018 (accessed February 13, 2018). It allows to specify coordinates of measurement points in the geographical coordinate system (latitude, longitude, altitude) either through a text file list, or mouse clicks on the virtual globe. The points and lidar beams are visualized on the 3D virtual globe and a simulation of the scans can be

demonstrated. The parameters for all specified lidars will be automatically generated. Lidar specific information about the measurement points, like pulse type to use, accumulation time and time to move between each measurement point, is defined in a settings tab for the whole scenario. Scenario files can be created and send to the systems.

Selected commands as specified in the open protocol RSComPro (Vasiljevic et al., 2013) of the WindScanner software have been implemented to increase the level of interaction of the software with the lidar systems. The most important commands

are "sending new scenarios" to a system and "starting" and "stopping" the measurements. The Java program will also receive current measurement data and display it upon request. Since synchronization of multiple systems is achieved by sending delays to the fastest system from the master computer, as described in Vasiljevic (2014), this task is also taken over by the Java program. Figure 3 shows a screenshot of the software in operation.

### 2.4 Adaptation of scanning scenarios to wind direction

With the previously described software it has also been realized that a scenario can be adapted according to an external input. For the given experiment, the goal was to measure in the wake of the wind turbine and thus, the measurement points should be adjusted to be downstream the wind turbine in the varying main wind direction. To achieve this, a rotation of the measurement points in the local coordinate system around the wind turbine position has been programmed, which is triggered by a new input of wind direction. The points are then rotated by the angular difference between the previous and the new wind direction. After

new measurement point locations have been computed, the lidars are stopped, the new scenarios are sent to the systems, and measurements are restarted (see Fig. 4). The time to restart the lidars with a new scenario can be as fast as 15 s, but occasionally also takes longer (< 1 min), if the initialisation of one of the system fails and needs to be repeated. Two different ways to obtain the wind direction where implemented:

1. VAD scans by lidar #3:

Lidar #3 is located approximately 300 m North-West of the wind turbine at almost equal height. In the cases of Westerly winds, which are of major interest, the air above the lidar is not affected by the wind turbine wake. Because of these boundary conditions, it was chosen to use this lidar for VAD scans to determine the inflow wind direction for the wind turbine in absence of any information from the wind turbine itself. In the planning software, the interval in which VAD scans are to be performed and the duration of the scan can be freely chosen. For this experiment it was set to be a

5 minutes scan every 30 minutes. With a minimum range gate distance of 100 m and an elevation angle of $75°$, the measurement height is at 97 m, which is 15 m above hub height of the turbine. After the 5 minutes of VAD, the wind speed and wind direction are calculated in real-time and the new wind direction is set in the software, triggering the adaptation of the scanning scenario.





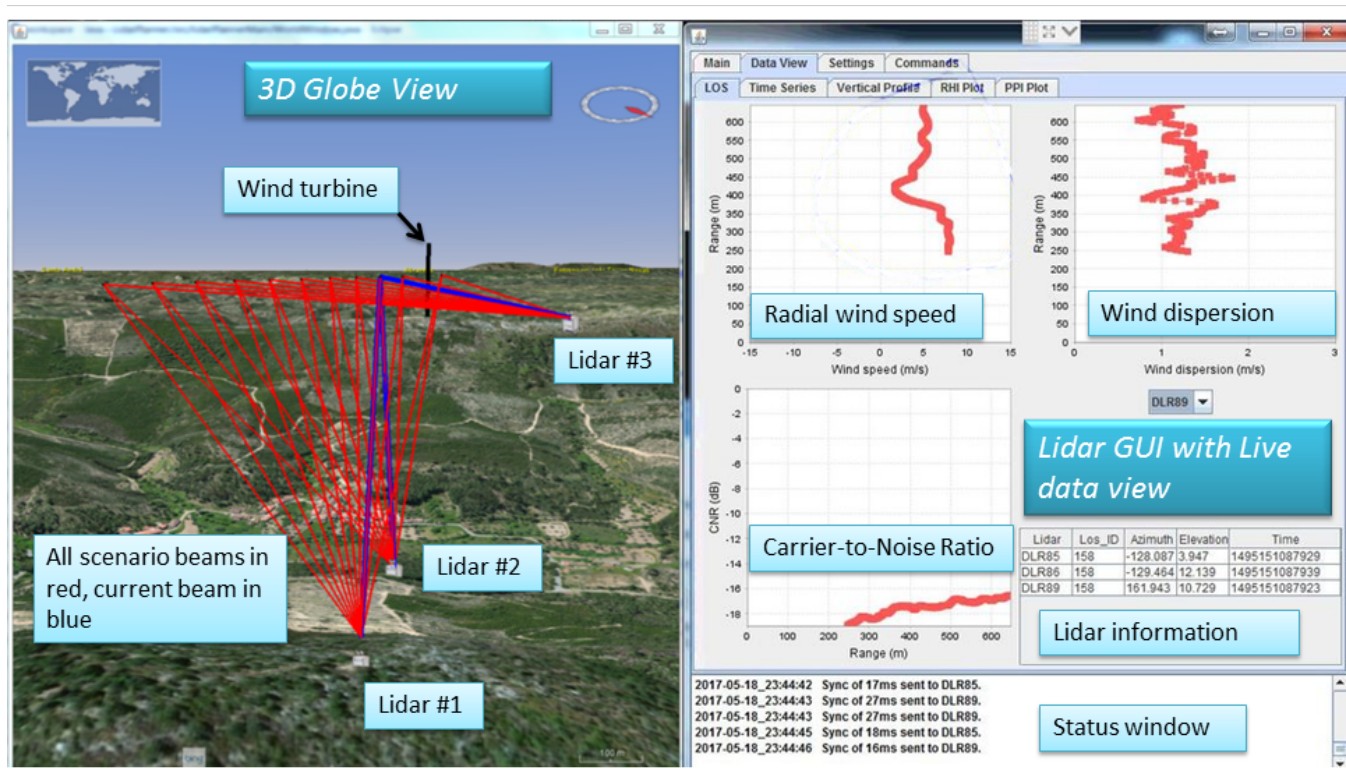

**Figure 3.** Example view of lidar planner software. On the left, the virtual globe with display of lidars and scanning scenario. On the right, the GUI for live data view. Other tabs in the program allow to change settings of the scenario, configure the automatic scenario adaption etc.




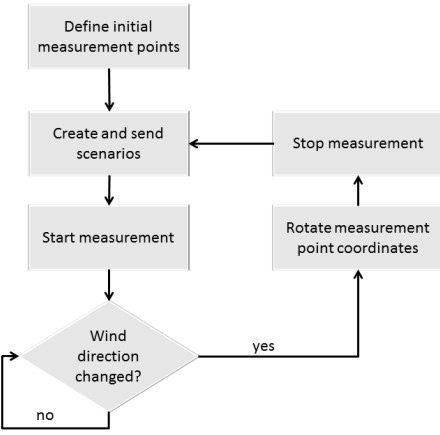

**Figure 4.** Flow diagram of the automatic adjustment of scanning scenarios.

2. Sonic anemometer measurements of meteorological mast 20/tse04:

   After three weeks in the IOP, i.e. after 19th of May, 5-minute average data of the meteorological masts was made available through an FTP server by NCAR in real-time. The closest meteorological mast to the wind turbine with a sonic anemometer at hub height (80 m above ground) is tower 20/tse04, which is depicted in Fig. 1. A Python-script was used to pull the actual wind direction every 30 minutes from the sonic anemometer at 80 m of tower 20/tse04 as NetCDF data and copy it into a local text file. This text file was monitored by the Java lidar program for changes: Every time a change in the file was detected, the new wind direction was pulled from the file, lidar scenarios adapted and lidar systems restarted.

Both methods are good examples how multiple instruments can be used together in a campaign to optimize the measurement strategy.

## 2.5 Measurement periods

During the IOP, the three DLR long-range lidars were mainly performing continuous vertical scans to capture all flow features over the valley. The experimental scanning with adapting scenarios was only carried out in selected periods with varying Westerly to Northerly winds of moderate strength. A list of the measurement periods is given in Tab. 2. For the analysis in this study, two subsets of these periods are picked in which the systems were all working fine and the wind direction showed some variation. The first period is from 17 May 2017, 17:00 UTC until 18 May 2017, 06:30 UTC, where the wind direction updates were retrieved from VAD scans of lidar #3. The second period is from 02 June 2017, 16:00 UTC until 02 June, 20:00 UTC, where tower 20/tse04 provided the wind direction updates.





| start time / UTC | end time / UTC | reference |
|---|---|---|
| 08 May 2017, 16:00 | 21:00 | VAD |
| 14 May 2017, 10:00 | 21:00 | VAD |
| **17 May 2017, 07:00** | **19 May 2017, 09:00** | **VAD** |
| 01 June 2017, 16:00 | 20:00 | tower 20/tse04 |
| **02 June 2017, 14:00** | **21:00** | **tower 20/tse04** |
| 03 June 2017, 13:00 | 15:00 | tower 20/tse04 |
| 04 June 2017, 10:00 | 20:00 | tower 20/tse04 |
| 06 June 2017, 09:00 | 11:00 | tower 20/tse04 |
| 08 June 2017, 11:00 | 14:00 | tower 20/tse04 |
| 14 June 2017, 12:00 | 18:00 | tower 20/tse04 |

**Table 2.** List of periods with adaptive multi-Doppler measurements. The column "reference" indicates if wind direction reference was obtained from VAD scans or meteorological tower 20/tse04. In bold are the periods that are looked at in detail in this study.

## 3 Methods

### 3.1 Velocity Azimuth Display

The velocity azimuth display (VAD) method was chosen to determine wind direction and wind speed with lidar #3. The scans were carried out with an elevation angle of $\varphi = 75°$, an angular speed of $10°$ s$^{-1}$ and a range gate distance of 100 m. The
5 calculation of wind speed and wind direction is done through a fit of the curve of radial wind speeds $v_r$ over azimuthal angle $\theta$ to Eq. 1.

$$v_r(\theta) = a\cos(\theta - \theta_0) + b, \tag{1}$$

with $a$ the amplitude of the sinusoidal fit, which translates to horizontal wind speed through $u = \frac{a}{\cos\varphi}$, and $b$ its offset, from which vertical wind speed $w = \frac{b}{\sin\varphi}$ can be calculated. Wind direction is the phase shift $\theta_0$ of the fit (see also Newman et al.,
2016). The calculation of horizontal wind speed through this method assumes a homogeneous and divergence-free wind field, which is doubtful in the given complex terrain, but will be evaluated for the wind direction estimation in this study in Sect. 4.

### 3.2 Multi-Doppler wind vector calculation

With multiple linearly independent radial wind speed measurements at the same point in space, the meteorological wind vector can be calculated. To calculate the three-dimensional wind vector, at least three independent radial wind speeds are necessary.
The relation between meteorological wind vector $[u, v, w]$ and radial wind speeds $[v_{r1}, v_{r2}, v_{r3}]$ can be described as:

$$\begin{bmatrix} v_{r1} \\ v_{r2} \\ v_{r3} \end{bmatrix} = \begin{bmatrix} \sin\theta_1\cos\varphi_1 & \cos\theta_1\cos\varphi_1 & \sin\varphi_1 \\ \sin\theta_2\cos\varphi_2 & \cos\theta_2\cos\varphi_2 & \sin\varphi_2 \\ \sin\theta_3\cos\varphi_3 & \cos\theta_3\cos\varphi_3 & \sin\varphi_3 \end{bmatrix} \begin{bmatrix} u \\ v \\ w \end{bmatrix}. \tag{2}$$





### 3.3 Wake center estimation

In this study, wake center positions are determined from single line-of-sight measurements of the three lidars. It is assumed
that the wake takes a shape as described in Trujillo et al. (2011), so that single line of sight measurements through the wake
center will manifest in a Gaussian-shaped velocity profile. Since the line-of-sight measurements have a certain elevation angle,
the vertical wind profile is superimposed on the Gaussian profile. Before fitting a Gaussian function as in Eq. 3 to the line of
sight measurement, it is therefore reduced by the linear trend.

$$v(x) = A_{\mathrm{g}} e^{-\frac{(x-b_{\mathrm{g}})^2}{2\sigma^2}} + d \quad , \tag{3}$$

where $A_{\mathrm{g}}$ is the amplitude of the Gauss-function and $\sigma$ is the standard deviation. The center of the fitted Gaussian function $b_g$
represents the wake center position along the lidar beam.

### 3.4 Wake models

In order to compare the results of the measurements to a well-known wake model, the Jensen-Park model is used in this study
according to: Jensen (1983); Peña et al. (2016):

$$1 - \frac{u(x)}{u_\infty} = \frac{1 - \sqrt{1 - C_t}}{(1 + k_w 2x/D)^2}, \tag{4}$$

with $u(x)$ the wind speed at distance $x$ to the WEC, $C_t$ the thrust coefficient of the wind turbine, whose exact value is unknown
in this study, but assumed to be 0.78. $k_w$ the wake decay coefficient and is defined as $k_w = 0.5/\log\left(\frac{h}{z_0}\right)$, with $z_0$ the roughness
length, which is set to 1 m for the case of a forest which is mostly covering the hills in Perdigão, and $h$ the hub height of the
wind turbine. $D$ is the rotor diameter and $u_\infty$ the wind speed of the inflow.
It is known to the authors that the Jensen-Park model is a very simplified engineering model, and the implied physics cannot
be assumed to provide realistic results for the given experiment, but it is a good reference, because it is well known to the
community.

## 4 Results

### 4.1 VAD scan and mast comparison

To evaluate the accuracy of the wind direction measurements by the VAD scan in complex terrain, the values that were measured
in real-time and used for the scenario control are compared to 10-minute averages of the sonic anemometer at 80 m of tower
20/tse04 in Fig. 5. It shows that wind direction trends agree well, but differences of up to 10-20 ° can be observed, with a mean
squared error (RMSE) of 14.04 °. The estimation of horizontal wind speed shows larger differences, especially during daytime,



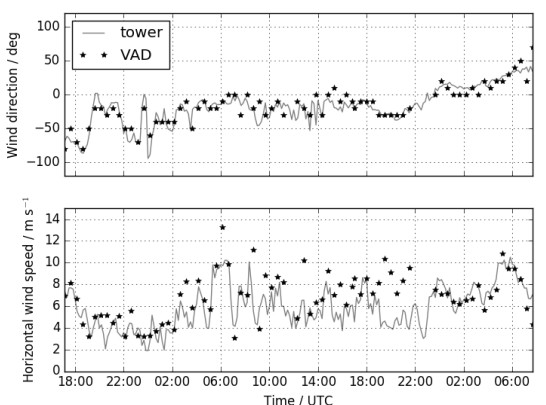
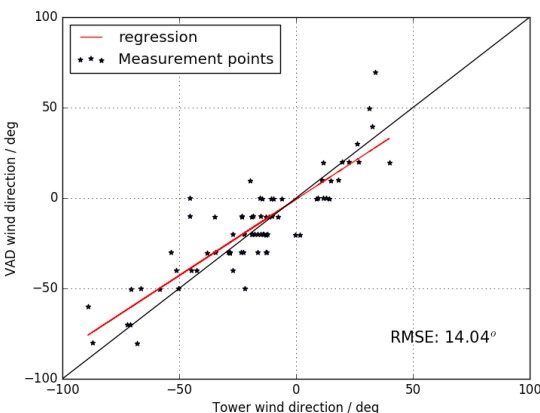

**Figure 5.** Comparison of lidar #3 VAD scans with sonic anemometer measurements at 80 m on tower 20/tse04 from 17th to 19th of May. On the left a comparison of the time series of wind direction and wind speed, and on the right a regression plot of wind directions.

which is likely due to convection and a vertical wind component that cannot be neglected and leads to an overestimation of the VAD measurement.

## 4.2 Synchronization accuracy

For the purpose of crossing beams of multiple lidars at the same point in time and space, high demands are made on the pointing

and timing accuracy of each system. In Vasiljevic (2014) it is explained in detail how hard target calibration of the pointing direction can be done, and how the timing synchronization based on a precise GPS clock and synchronization commands of the master computer is realized in the WindScanner system. The accuracy of the hard target calibration of the azimuth and elevation angle offsets is dependent on the accuracy of the location measurement of the lidars and the targets. For the DLR lidars in the Perdigão 2017 campaign, locations were first measured with a single-frequency GPS receiver in precise-point-

positioning (PPP) mode, which allows accuracies on the order of 1-2 m. After three weeks of the campaign, a dual-frequency GPS receiver with centimeter-accuracy was available, and calibration was repeated. It was found that the offsets with the more accurate location measurement differed up to 0.6 ° from the original calibration. This error is not uniquely due to the wrong position measurement, but also due to maladjustment because of slight movements of the lidar on a non-solid ground. An error of 0.6 ° for lidar #1 leads to a spatial offset of the measurement point of approximately 15 m at the location of the WEC and has

to be taken into account. The error is however of the same magnitude as the physical resolution of the lidar, so it is considered acceptable for further analysis. The large error applies to the measurements from 17th to 19th May. Measurements on 02 June were done with the more accurate readjustment.

Usually, the WindScanner software allows timing synchronization down to 10 ms of all systems, which was demonstrated in a number of campaigns with a development version of the software (Vasiljevic et al., 2016). Because of a bug in the commercial

version of the WCS that miscalculated scanner movement times depending on the acceleration and deceleration periods, timing





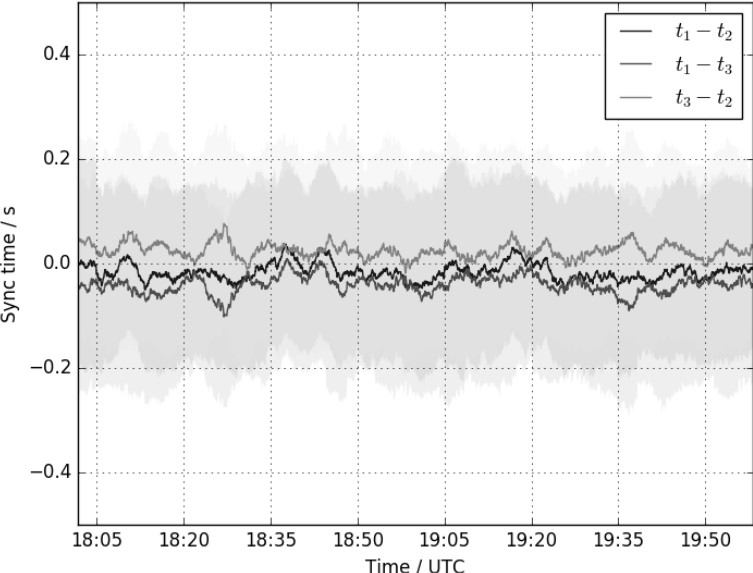

**Figure 6.** Time difference between lidar #1 ($t_1$), lidar #2 ($t_2$) and lidar #3 ($t_3$) for an exemplary period on 02 June. The lines show the moving average synchronization error over 100 scanning points and the shaded area is the standard deviation respectively.

was less accurate in the presented experiment. Fig. 6 shows the time difference between the three systems for each measurement point in a two-hour period on 02 June. It shows that the standard deviation of the timing errors mostly stays below 0.2 s, which is an acceptable error for the given experiment where mean wind speeds are regarded but is critical for turbulence measurements.

## 4.3 Wake measurements

### 4.3.1 Wind speed deficit

With the described measurement strategy, wind speed deficits can be calculated downstream the turbine in main wind direction in dependency of the distance to the rotor. Fig. 7 and 8 show the results as horizontal wind speed measurement and wind speed deficit, normalized to the upstream wind speed $v_\infty$ for the two days of interest. For all cases, $v_\infty$ is the extrapolated wind speed at hub height from measurements of the upstream tower 37/rsw06, with sonic anemometer measurements up to 60 m. For the analysis, only 30-minute averages with inflow wind speeds of more than 5 m s$^{-1}$ are used.

The results show that the actual wind speed deficit is larger than predicted by the Jensen-Park model in both cases and for the majority of 30-minute averages. Some of the 30-minute averages show a fast decay of the wind speed deficit, especially on 17 May. In these cases, the alignment of the measurement points with the wake center was too far off, which has the effect





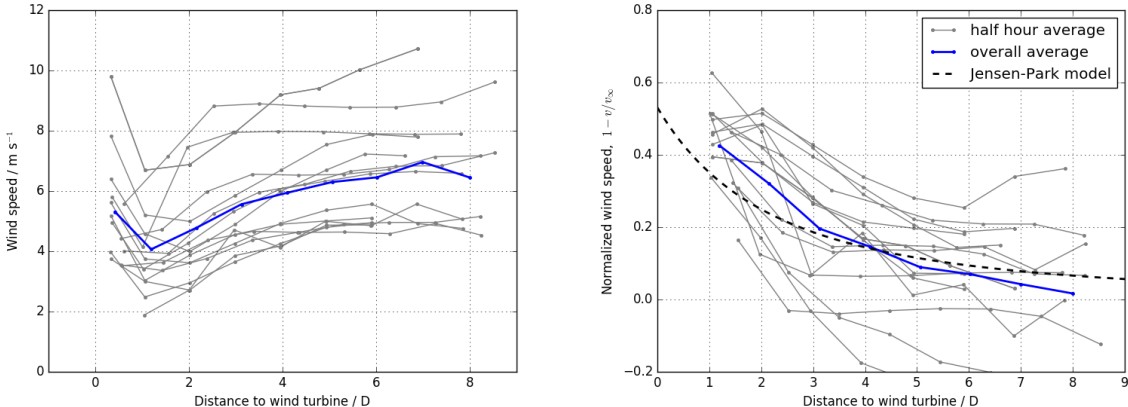

**Figure 7.** Multi-Doppler wind speed measurements in dependency of distance to the WEC (left), and wind speed deficit normalized to undisturbed wind speed measurement on tower 20/tse04 at 80 m on 17 May 2017. Gray lines are averages of a 30-minute period, the thick blue line is the overall average.

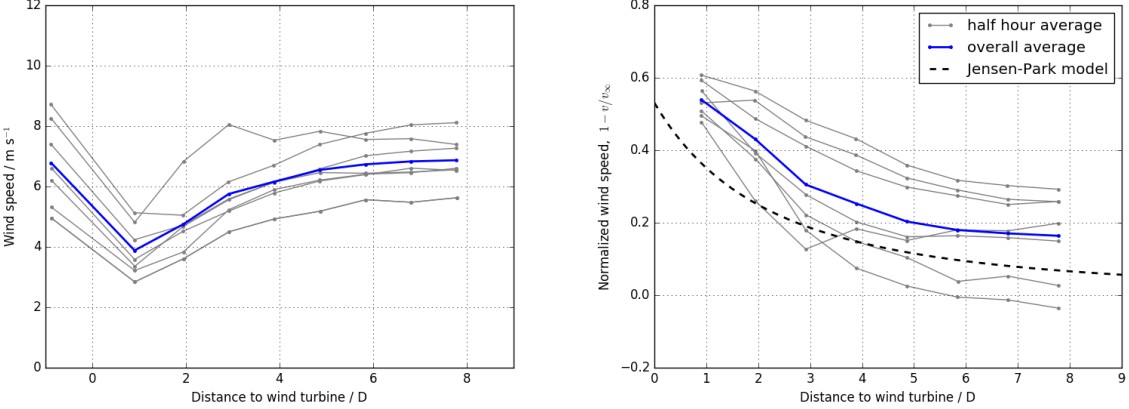

**Figure 8.** Multi-Doppler wind speed measurements in dependency of distance to the WEC (left), and wind speed deficit normalized to a measurement point one rotor diameter upstream on 02 June 2017. Gray lines are averages of a 30-minute period, the thick blue line is the overall average.

that measurement points at far distances to the WEC are outside the wake and thus the wind speed deficit gets small and is not representing the wake any more. This situation will be evaluated in the next section.





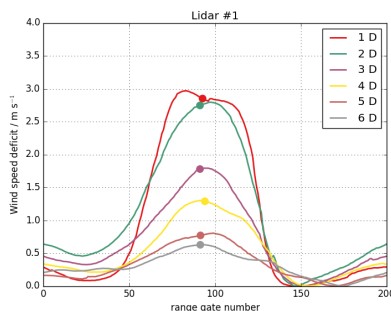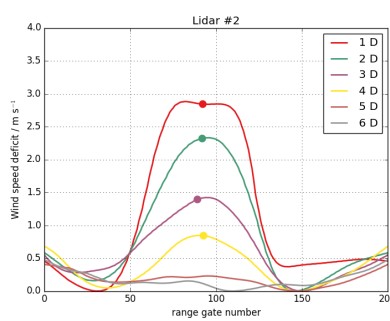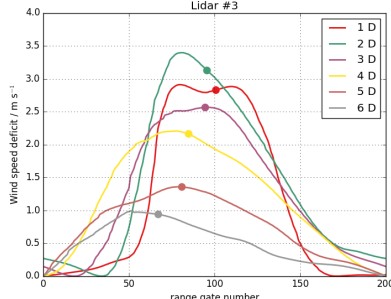

**Figure 9.** Averaged line-of-sight measurements through the wake by the three lidar systems in different distances to the WEC. The dots show the estimated wake center positions by a Gaussian fit.

### 4.3.2 Wake center position

With additional range gates before and after the crossing point of the lidar beams, a spatial resolution of the wake shape and its center position can be achieved. On 17 May, the extra range gates were only placed 50 m before and behind the lidar beam crossing point, with a resolution of 1 m. This showed to be not enough for a proper fit of a theoretical wake profile to the data. In later measurements, as on 02 June, range gates were placed 200 m before and behind the crossing point with a resolution of 2 m.

Fig. 9 shows averaged line-of-sight measurements through the wake for one half-hour period. For lidars #1 and #2 the estimated wake center is quite close to the center of the range gate (although never perfectly matching it) with a trend of a bigger displacement with increasing distance to the WEC. For lidar #3 the mismatch is very obvious, because this lidar cuts the wake with the highest elevation angle whereas the wake flow follows the terrain downslope and the wake center appears in range gates closer to the lidar than the center range gate.

The misalignment error can be quantified as the mismatch between estimated wake center position and the position where the lidar beams cross, as depicted for each half-hour measurement period against the distance to the WEC in Fig. 10. The offset is obviously large and should be reduced in future experiments by a better prediction of the wake center, taking into account the wind turbine yaw angle if possible and reducing the update cycle to less than 30 minutes.

If, instead of the wind speeds at the lidar beam crossing point, the minimum wind speeds of the 400 m long range with measurement points along the line-of-sights of the lidar beams are used for the wind speed retrieval on 02 June, an even larger wind speed deficit is found (see Fig. 11). Of course, this will also not be the real wind speed deficit in the center of the wake, because not all three beams cut the wake at its center. This however means that the true wind speed deficit in the wake is even larger and thus, engineering models like the Jensen-Park model cannot describe the true energy loss in wakes satisfactorily.



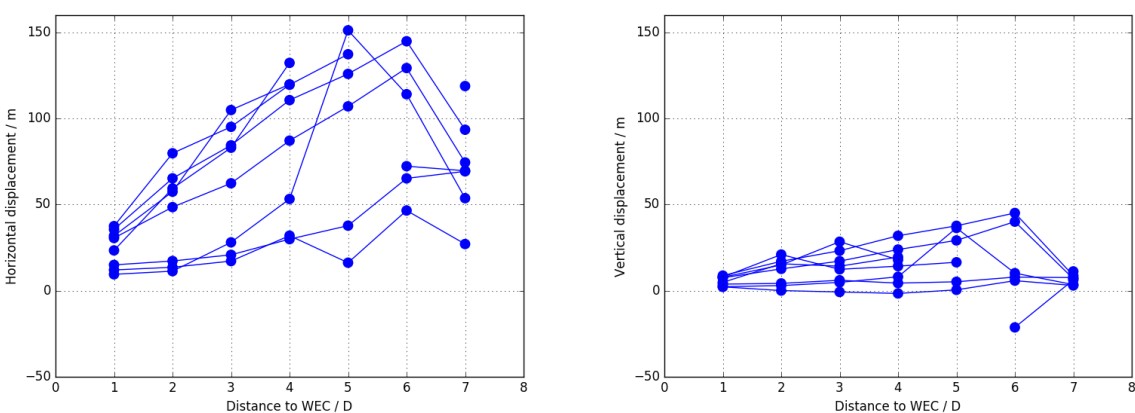

**Figure 10.** Averaged misalignment of the determined wake center to the beam crossing measurement point.

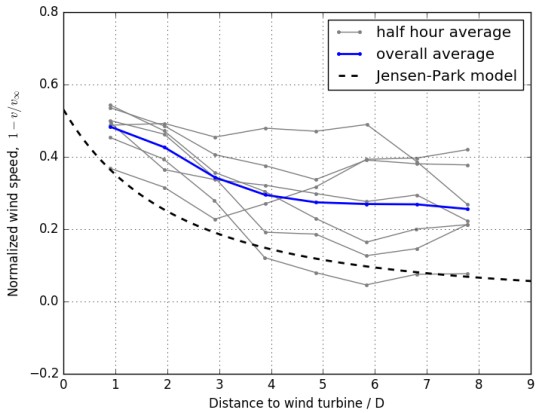

**Figure 11.** Wind speed deficit behind the WEC, if minimal wind speeds of line-of-sight measurements of the three lidars are used for wind speed retrieval.



## 5 Conclusions

With this study it is shown for the first time that long-range lidar systems - upgraded with the WindScanner software - can be used in a flexible way with automatically adjusting scanning trajectories. In the context of wind turbine wake measurements this setup is a powerful tool because it enables a continuous monitoring of the wake in a multi-Doppler mode with ground-based lidars, whereas a static configuration is only able to provide wake data in short periods of time. It is shown that the simple Jensen-Park engineering model for the wind speed deficit in the wake is underestimating the real wind speed deficit for the presented case studies of the Perdigão 2017 experiment. Measured wind speed deficits are at least 10% higher than the model predicts for distances up to eight rotor diameters downstream the WEC.

In ten measurement periods, the method has been tested and developed. A number of lessons learned are given here to summarize the potential and limitations of the measurement strategy:

- The slow synchronization that has been described in Sect. 4.2 has already been fixed in the WindScanner software. In future experiments it will thus be possible to scan much faster, achieving higher temporal resolutions and better synchronization of the lidars which will hence allow the resolution of turbulence statistics.

- The wind directions used for the scan adaptation in this study was first based on VAD scans of one of the lidars and later on sonic anemometer measurements of a tower close to the WEC. It is shown that both strategies work in general but using an external source for the wind direction increases the availability. Especially in complex terrain, VAD scans have significant errors compared to sonic anemometer measurements. The combination of tower data and lidar measurements is a good example towards a "smart" experimental setup, which in this case was possible because of the extraordinary infrastructure in Perdigão. In experiments with less infrastructure and less complex terrain, the VAD-method can still be a good alternative. Higher update rates than the 30 minutes that were used in this experiment are necessary if wind conditions and yaw angle of the WEC change as quickly as they did in Perdigão.

- Large improvements are expected if the wind direction measurement of the wind turbine itself, or even better its measured yaw angle could be used for control of the lidars. Another option would be to track the wake in real time and use this information for control of lidar scans but this would require more independent instrumentation.

- A reduction of the time for reconfiguring the scanning trajectories will further increase the data availability.

- Since the wake will always deviate slightly from the predicted location, multiple range gates before and after the determined center position for each line of sight measurement are helpful to determine the wake center in post-processing and also allow some analysis of wake extension. The range gates should be close to each other, but the range should also be large enough to cover the whole width of the wake and allow automatic wake center detection with suitable fitting functions. Range gates 200 m (or 2.5 rotor diameter) before and behind the determined range gate center, with a separation of 2 m have proven to be suitable for the given WEC.



- An extra measurement point should be set at a minimum of 2.5 rotor diameters upstream for a better estimation of the free stream velocity $v_\infty$.

A next step towards better understanding of wake dynamics is to apply the lessons learned from this case study to a long-term campaign which focusses on this measurement strategy and will allow to obtain better statistics of the wake characteristics in
different atmospheric conditions. Currently, a research wind park is being planned in Northern Germany by the DLR and its partners which will be equipped with scientific instrumentation on all levels from rotor blade, nacelle, tower and foundation to the atmosphere in near and far field, including the systems presented in this study. It is evident that this facility will be an ideal platform to perform dedicated experiments in flat terrain in order to get more basic understanding of wake physics without the effects induced by complex terrain. With the dataset that was obtained in the Perdigão 2017 experiment, a good basis for
research in the field of interaction of a WEC with the atmosphere in complex terrain is given. In the future, numerical models on different scales will need to be compared to and validated with the dataset.

*Data availability.* Data of all instruments that were used in this study is stored on three mirrored servers owned by DTU, University of Porto and the NCAR Earth Observing Laboratory (EOL) respectively. The data is publicly available through dedicated web portals of the University of Porto (https://windsp.fe.up.pt/) and EOL (http://data.eol.ucar.edu/master_list/?project=PERDIGAO).

*Acknowledgements.* We want to thank José Palma, University of Porto and José Caros Matos and the INEGI team for the local organization and tireless work in order to make this experiment a success. We acknowledge all the hard work of the DTU and NCAR staff to provide large parts of the hardware and software infrastructure available at Perdigão and in particular Steve Oncley for providing real-time access to the tower data. We appreciate the hospitality and help we received from the municipality of Alvaiade and Vila Velha de Rodão throughout the campaign.
This work was performed within projects LIPS and DFWind, both funded by the Federal Ministry of Economy and Energy on the basis of a resolution of the German Bundestag under the contract numbers 0325518 and 0325936A, respectively.





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
