# Peer review of "Wind Turbine Wake Measurements with Automatically Adjusting Scanning Trajectories in a Multi-Doppler Lidar Setup"

_Atmospheric Measurement Techniques, 2018_

## Referee Comment (RC1) · Anonymous Referee #1 · 23 Mar 2018

Review of the manuscript amt-2018-55 entitled: "Wind turbine wake measurements with automatically adjusting scanning trajectories in a multi-Doppler lidar setup", by N. Wildmann, N. Vasiljevic, T. Gerz.

This manuscript presents a test from the Perdigao field campaign using fixed-point triple-Doppler lidar measurements. According to the occurred wind direction, one lidar scanned roughly from an upstream position, while the other to lidars scanned from the same side of the wake.

The scanning setup is far to be an ideal configuration for 3D velocity retrieval. As I suggest below at comment 2, a quantification of the expected accuracy on the 3D velocity

retrieval should be provided. This analysis has even more relevance considering the complex topography of the Perdigao site.

The results presented consist in time-averaged streamwise wind velocity at fixed points for various downstream locations. No information is provided about the other two velocity components. No information is provided about the possible relative position of the measurement points within the wake.

The time-averaged streamwise velocity is then compared to the prediction of the Jensen-Park model. In my opinion, this analysis is not really pertinent; indeed, I don't see any reason why a space-averaged velocity deficit obtained from a top-hat model should match with a time-averaged fixed point measurement.

Then, time-averaged line-of-sight measurements are provided. As mentioned before, two lidars measured roughly transversally to the wind direction; thus, I understand the authors detected a velocity deficit; however, I am skeptical about the measurement accuracy of these measurements considering the large azimuthal angle between laser beam and wind direction.

I think that the highest potential of these measurements has not been exploited yet, namely analysis of 3D turbulence within a wake. In contrast, the analysis of the time-averaged velocity field adds little value to our understanding on wind turbine wakes. Detailed comments are provided below, which might help for a revised manuscript.

1. P1L16: wind energy converter is something different than a wind turbine? This name sounds a bit exotic. Consider to name it simply wind turbines;

2. Fig. 2. From this figure, it seems that the scanning scenario used to retrieve 3D velocity components is far to be ideal. One lidar laser beam is roughly in the mean wind direction, while the other two lidars are practically orthogonal. Can you use the method proposed by Debnath et al. 2017, Atmos. Meas. Tech. 10, 431-444, (Table 5) to quantify expected accuracy for the 3D lidar retrieval? Maybe this analysis can be

added to Sect. 3.2.

3. Sect. 3.3. This section needs a broader description and more details. For instance, which lidar(s) did you use to detect the wake center? Did you analyze simply the radial velocity? Lidars 1 and 3 are practically orthogonal to the typical wind direction, how did you characterize the wake velocity field from these data?

4. P10L16 In the same line you use WEC and turbine, but I guess you mean the same thing, namely the wind turbine. Consider to call it simply wind turbine.

5. P10L18: Calculating the expansion factor, kw, from log law has no sense for such complex topography. At least, mention this comment and explain that you cannot have better predictions with the available data.

6. Sect. 4.1: I guess that the poor accuracy of the VAD can be ascribed to the complex topography under investigation. You can add this comment in the text. Please report, slope, bias and r-square value of the linear regression between met-tower and VAD for both wind speed and direction.

7. P12L12 "The results show that the actual wind speed deficit is larger than predicted by the Jensen-Park model in both cases and for the majority of 30-minute averages." Why did you expect that that a top-hat model, such as the Jensen-Park model, could predict the same velocity deficit as for point-wise measurements? Likewise, for P14L20.

8. Sect. 4.3.2 and Fig. 9: Lidars 1 and 2 have a large azimuthal angle with respect to the mean wind direction, let's say larger than 45 degree. I don't think that the analysis of the radial velocity may enable accurate estimates of the velocity deficit.

---

## Referee Comment (RC2) · Anonymous Referee #2 · 23 Mar 2018

This manuscript describes the method how to measure fluctuating wakes behind a wind turbine with a system of three synchronised wind lidars.

The manuscripts mainly focuses on the measurement method which is very much appropriate for the chosen journal.

Nevertheless, the interpretation of the sample measurements should be scientifically sound and interesting for a broader audience.

I have two major points.

I do not fully understand why two lidars have been sited close to each other and the

third one further away. I would expect the greatest flexibility in measuring fluctuating wakes when having the lidars in a sort of an equally-sided triangle around the expected measurement volume.

I do not fully understand why the results have been discussed in terms of the Jensen Park model. This model has been developed for flat terrain and essentially neutral thermal stratification. The formula given for kw essentially says that kw is equal to turbulence intensity. Thus, turbulence measurements could be used to test the validity of the calculation of kw.

For the assessment of the samples shown, it would be really interesting to learn something about thermal stability and overall turbulence intensity during these measurement periods. Maybe, this would be the clue to the overestimation or underestimation of the wake.

---

## Author Comment (AC1) · 18 May 2018

**1 Author response**

We want to thank the two anonymous reviewers for their valuable feedback and valid points of criticism to our manuscript describing automatically adjusting scanning trajectories in a multi-Doppler lidar setup. In the presented study, we wanted to focus on the atmospheric measurement technology, showing the possibilities with state-of-the-art scanning Doppler lidar systems in a large experimental setup and especially focussing on wind turbine wake measurements. A main point of criticism of both reviewers is

that the behaviour of the wake has not been analysed in all details with regard to atmospheric stability, turbulence in the wake, lateral wind speeds in the wake, etc. We are aware that all these topics are highly relevant and we also believe that the presented technology will help to study these processes in the future, but in the presented manuscript we want to emphasize the measurement technology and measurement strategy and decided to limit the scope of this paper to the most basic analysis of mean wind speed deficit for two reasons. First, the database is comparatively small, because the technique to adapt the scanning trajectories has been implemented for the first time and a much longer time during the Perdigão experiment was used for continuous RHI scans at a fixed angle. The results of these scans are presented in a different publication. Second, during the Perdigão experiment, the DLR WindScanners were suffering from a software bug, as described in the manuscript and could only be operated comparatively slowly, which limits the possibility to analyse turbulence to only the large scales. In the direct responses to the reviewer comments we will elaborate in detail on the mentioned issues and suggest to add some additional analysis to the revised manuscript, always highlighting the constraints of the dataset. Despite the constraints and limitations of the dataset, we still believe that the technique of measurements, the proof-of-concept and the results of the mean wind speed deficit is of high value for the scientific community and suitable for publication in AMT. It will hopefully foster future long-term campaigns with similar scanning strategies and smart layouts of experiments.

**1.1 RC1, General Comments**

• The scanning setup is far to be an ideal configuration for 3D velocity retrieval. As I suggest below at comment 2, a quantification of the expected accuracy on the 3D velocity retrieval should be provided. This analysis has even more relevance considering the complex topography of the Perdigao site.

As decribed in Sect. 2.2, the lidar siting was done for a different primary scanning

strategy. Due to logistical constraints, compromises had to be made for the location of the lidars. However, the geometry of lidar beams in the area of interest still achieves reasonable uncertainties for the primary area of interest as we want to show here. The equation

$$\begin{bmatrix} v_{r1} \\ v_{r2} \\ v_{r3} \end{bmatrix} = \begin{bmatrix} \sin\theta_1 \cos\varphi_1 & \cos\theta_1 \cos\varphi_1 & \sin\varphi_1 \\ \sin\theta_2 \cos\varphi_2 & \cos\theta_2 \cos\varphi_2 & \sin\varphi_2 \\ \sin\theta_3 \cos\varphi_3 & \cos\theta_3 \cos\varphi_3 & \sin\varphi_3 \end{bmatrix} \begin{bmatrix} u \\ v \\ w \end{bmatrix} = \vec{M}\vec{u}$$
(1)

and its solution for the meteorological wind vector  $\vec{u}$

$$\vec{u} = \vec{M}^{-1} \vec{v_r} \tag{2}$$

describe the relationship between  $\vec{u}$  and radial velocities  $\vec{v_r}$  of three lidar systems. The uncertainty of the retrieval of the wind velocity component u (and equivalently for v and w) that is introduced by the geometrical arrangement of the lidars can be calculated by propagating the uncertainty of each lidar's radial wind speed uncertainty through Eq. 2:

$$\epsilon_{u} = \sqrt{\sum_{j=1}^{3} \left(\frac{\partial u}{\partial v_{\mathsf{r},\mathsf{j}}}\right)^{2} (\epsilon_{j})^{2}} \tag{3}$$

as it has for example been described for dual-Doppler applications by Hill et al. (2010). For a common uncertainty  $\epsilon$  of all lidars, the equation is equivalent to a multiplication of the  $\ell^2$ -norm of the column vector of  $\vec{M}^{-1}$  with  $\epsilon$  and thus equivalent to the method described in Debnath et al. (2017). Pauscher et al. (2016) has shown for the same type of instruments as they are used in this study (Leosphere Windcube 200S with WindScanner software), that the uncertainty of radial wind speeds is on the order of 0.1 m s-1 when compared to a sonic anemometer. If this value is used as the input uncertainty  $\epsilon$  for Eq. 3, even the uncertainty of the

w-component is approximately 1 m s-1 in the sector of interest. It was planned to run the adaptive scanning strategies for wind directions ranging from 240° to 50° in clockwise direction. For the two analysed periods, only wind directions between approximately 290° and 360° occured. Figure 1 shows a colormap of the uncertainties at measurement height (hub height) in a radius of ten rotor diameters around the wind turbine and depicts the target sector and the actual measurement points. It is evident that in the direction where lidar locations are almost collinear, the uncertainties are extremely high, but in the sector which is relevant for westerly and north-westerly winds, as they are found in the case studies, the uncertainties are lowest. In Wildmann et al. (2018) we showed a study comparing radial wind speeds measured in Perdigão with a 100 m mast in the valley. The results of this comparison are on the order of 0.4 m s-1, which is likely due to the very complex flow in that area and the increased uncertainty due to the spatial distance between sonic anemometer and lidar measurement volume in the experimental setup. However, calculating the uncertainty map for this input uncertainty still shows horizontal wind speed component uncertainties below 1 m s-1 whereas the vertical wind component becomes very uncertain (see Fig.2). It is also for this reason that the analysis in the manuscript has been done with horizontal wind speeds only.

We will add this analysis to Sect. 3.2 of the revised manuscript.

• The results presented consist in time-averaged streamwise wind velocity at fixed points for various downstream locations. No information is provided about the other two velocity components. No information is provided about the possible relative position of the measurement points within the wake.

The streamwise velocity component is the most relevant component for power production of the wind turbine. An analysis of vertical and lateral wind speed component which is certainly very relevant to the process understanding of the wake requires an in-depths analysis and a larger database which is outside the scope of this study. Relative positions of the measurement points within the wake are discussed in Sect. 4.3.2 in the manuscript.

- The time-averaged streamwise velocity is then compared to the prediction of the Jensen-Park model. In my opinion, this analysis is not really pertinent; indeed, I don't see any reason why a space-averaged velocity deficit obtained from a top-hat model should match with a time-averaged fixed point measurement. The Jensen-Park model is presented as a reference for the reader in order to have some connection to previous studies and existing wake models. It is very clearly stated in the text that a good fit is not expected, but it is nevertheless interesting to see in which direction and to what degree the measurement spresented in the revised manuscript as described in the answer to review #2 by using a turbulence intensity based wake decay coefficient  $k_w$ .
- Then, time-averaged line-of-sight measurements are provided. As mentioned before, two lidars measured roughly transversally to the wind direction; thus, I understand the authors detected a velocity deficit; however, I am skeptical about the measurement accuracy of these measurements considering the large azimuthal angle between laser beam and wind direction.

The line-of-sight measurements are used to detect the position of the wake center and not to quantify a velocity deficit. Velocity deficits are calculated with the combined multi-Doppler solution of all instruments.

• I think that the highest potential of these measurements has not been exploited yet, namely analysis of 3D turbulence within a wake. In contrast, the analysis of the timeaveraged velocity field adds little value to our understanding on wind turbine wakes. Detailed comments are provided below, which might help for a revised manuscript.

We decided to not include turbulence measurements, especially an analysis of

the 3D structure of the turbulence to this manuscript due to the reduced sampling rate and time synchronization accuracy as described in the manuscript. We believe that with a WindScanner software that allows faster scans with 10 ms accuracy, this scanning strategy is perfectly suitable for extensive turbulence studies. In this study, the sampling rate at every measurement point is only about 1/20 s which means that at a mean wind speed of 5 m s-1 and according to the Nyquist theorem, only eddies larger than 240 m are sampled. It is however possible to calculate turbulence intensity at the triple-Doppler measurement points as  $I = \frac{\sigma_u}{\overline{u}}$ . Figure 3 shows the result of this analysis in comparison to measurements at tower 37/rsw06 for the sonic anemometer at 60 m. The sonic anemometer *I* is calculated in the same way, but standard deviation and mean value are calculated for 10-min periods with a sampling rate of 20 Hz. It shows that in close vicinity to the wind turbine, *I* is twice as high as in the undisturbed atmosphere, but at 8 D downstream it is back to the background level in almost all cases. We will add a short section after Sect. 4.3.1 in the revised manuscript.

**1.2 RC1, Specific Comments**

- P1L16: wind energy converter is something different than a wind turbine? This name sounds a bit exotic. Consider to name it simply wind turbines; Wind energy converter is a correct technical term, that is also widely used in the scientific community. We chose to use both terms (WEC and wind turbine) synonymously to make the text less repetitive. We will change everything to "wind turbine" in a revised manuscript.
- Fig. 2. From this figure, it seems that the scanning scenario used to retrieve 3D velocity components is far to be ideal. One lidar laser beam is roughly in the mean wind direction, while the other two lidars are practically orthogonal. Can you use the method proposed by Debnath et al. 2017, Atmos. Meas. Tech. 10,

431-444, (Table 5) to quantify expected accuracy for the 3D lidar retrieval? Maybe this analysis can be added to Sect. 3.2.

We agree that we should add a section about the uncertainties from the laser beam geometry and have elaborated about this above as response to the general comments. Since we focus on measurements in the wind direction range between 240° and 50°, we believe that a reasonable accuracy is achieved. The uncertainty analysis will be added to a revised manuscript in Sect. 3.2.

- 3. Sect. 3.3. This section needs a broader description and more details. For instance, which lidar(s) did you use to detect the wake center? Did you analyze simply the radial velocity? Lidars 1 and 3 are practically orthogonal to the typical wind direction, how did you characterize the wake velocity field from these data? To estimate the wake center along the line-of-sight of the lidar beam, it is not necessary to obtain the absolute value of wind velocity, the shape of the radial velocities can be fit to a Gaussian function and the center of this fit serves as an estimate of the wake center. Even a cut through the wake at a slanted angle will result in typical Gaussian shaped wake profiles as shown in Fig. 9 of the manuscript.
- 4. P10L16 In the same line you use WEC and turbine, but I guess you mean the same thing, namely the wind turbine. Consider to call it simply wind turbine. see above
- 5. P10L18: Calculating the expansion factor, kw, from log law has no sense for such complex topography. At least, mention this comment and explain that you cannot have better predictions with the available data.

We will clarify this in the revised manuscript. In response to the comments of reviewer #2 we will change the estimation of  $k_w$  to a value based on turbulence intensity.

6. Sect. 4.1: I guess that the poor accuracy of the VAD can be ascribed to the complex topography under investigation. You can add this comment in the text. Please report, slope, bias and r-square value of the linear regression between met-tower and VAD for both wind speed and direction.

The comment about the complex terrain as a root cause for errors of the VAD method is mentioned both in Sect. 4.1 and the conclusion. It is however also noticeable that the errors are larger during daytime, which we attribute to the convective regime and have mentioned so in the text. Since wind speed measurements by the VAD are not really essential for the presented experiment we would omit to also show the regression plot of this variable in the revised manuscript. We will show it here and also mention the slope, bias and r-square in the plot in the revised manuscript (see Fig. 4). Obviously, the wind speed regression is strongly biased by the large errors of wind speed measurement during the convective daytime. It does not affect the wind direction as strongly. A more detailed analysis - though certainly interesting - is not part of this study and not really relevant here. We believe that it is important to show that three lidars alone can be used to design an adaptive scanning strategy and possible future experiments in flat terrain without measurement masts can profit a lot by the technique.

7. P12L12 "The results show that the actual wind speed deficit is larger than predicted by the Jensen-Park model in both cases and for the majority of 30-minute averages." Why did you expect that that a top-hat model, such as the Jensen-Park model, could predict the same velocity deficit as for point-wise measurements? Likewise, for P14L20.

We agree that the wording of this sentence is unfortunate. In Sect. 3.4 we already mention that this model cannot be assumed to predict the same velocity deficit as the measured velocity deficit, it is in our opinion still a good reference for the reader to see how wakes of this type of wind turbine are modelled in flat terrain and how the results of a measurement campaign in complex terrain can differ.

8. Sect. 4.3.2 and Fig. 9: Lidars 1 and 2 have a large azimuthal angle with respect to the mean wind direction, let's say larger than 45 degree. I don't think that the analysis of the radial velocity may enable accurate estimates of the velocity deficit.

It was not the goal to estimate velocity deficits trough the radial velocities. The purpose is to detect the center location of the wake in order to estimate the triple-Doppler measurement location within the wake.

**References**

- Debnath, M., lungo, G. V., Ashton, R., Brewer, W. A., Choukulkar, A., Delgado, R., Lundquist, J. K., Shaw, W. J., Wilczak, J. M., and Wolfe, D.: Vertical profiles of the 3-D wind velocity retrieved from multiple wind lidars performing triple range-height-indicator scans, Atmospheric Measurement Techniques, 10, 431–444, doi:10.5194/amt-10-431-2017, https: //www.atmos-meas-tech.net/10/431/2017/, 2017.
- Hill, M., Calhoun, R., Fernando, H. J. S., Wieser, A., Dörnbrack, A., Weissmann, M., Mayr, G., and Newsom, R.: Coplanar Doppler Lidar Retrieval of Rotors from T-REX, Journal of the Atmospheric Sciences, 67, 713–729, doi:10.1175/2009JAS3016.1, http://dx.doi.org/10.1175/ 2009JAS3016.1, 2010.
- Menke, R., Vasiljević, N., Hansen, K., Hahmann, A., and Mann, J.: Does the wind turbine wake follow the topography? A multi-lidar study in complex terrain, Wind Energy Sci., submitted, 2018.
- Pauscher, L., Vasiljevic, N., Callies, D., Lea, G., Mann, J., Klaas, T., Hieronimus, J., Gottschall, J., Schwesig, A., Kühn, M., and Courtney, M.: An Inter-Comparison Study of Multi- and DBS Lidar Measurements in Complex Terrain, Remote Sensing, 8, 782, doi:10.3390/rs8090782, http://www.mdpi.com/2072-4292/8/9/782, 2016.
- Peña, A., Réthoré, P.-E., and van der Laan, M. P.: On the application of the Jensen wake model using a turbulence-dependent wake decay coefficient: the Sexbierum case, Wind Energy, 19, 763–776, doi:10.1002/we.1863, we.1863, 2016.

Vasiljević, N., L. M. Palma, J. M., Angelou, N., Carlos Matos, J., Menke, R., Lea, G., Mann, J., Courtney, M., Frölen Ribeiro, L., and M. G.

**C9**

C. Gomes, V. M.: Perdigão 2015: methodology for atmospheric multi-Doppler lidar experiments, Atmospheric Measurement Techniques, 10, 3463–3483, doi: 10.5194/amt-10-3463-2017, 2017.

Wildmann, N., Kigle, S., and Gerz, T.: Coplanar lidar measurement of a single wind energy converter wake in distinct atmospheric stability regimes at the Perdigão 2017 experiment., Journal of Physics: Conference Series, doi:submitted, 2018.

**Fig. 1.** Map of uncertainties at hub height in a radius of  $10 \sim D$  around the wind turbine for  $u^{0}, v^{0}$  and  $w^{0} = 0.1$ .

Fig. 2. Same as Fig. ${\sim}1$  but for a radial wind speed uncertainty <code>epsilon=0.4</code>.

---

## Author Comment (AC2) · 18 May 2018

**1   Author response**

We want to thank the two anonymous reviewers for their valuable feedback and valid points of criticism to our manuscript describing automatically adjusting scanning trajectories in a multi-Doppler lidar setup. In the presented study, we wanted to focus on the atmospheric measurement technology, showing the possibilities with state-of-the-art scanning Doppler lidar systems in a large experimental setup and especially focussing on wind turbine wake measurements.  A main point of criticism of both reviewers is

that the behaviour of the wake has not been analysed in all details with regard to at-mospheric stability, turbulence in the wake, lateral wind speeds in the wake, etc. We are aware that all these topics are highly relevant and we also believe that the presented technology will help to study these processes in the future, but in the presented manuscript we want to emphasize the measurement technology and measurement strategy and decided to limit the scope of this paper to the most basic analysis of mean wind speed deficit for two reasons. First, the database is comparatively small, because the technique to adapt the scanning trajectories has been implemented for the first time and a much longer time during the Perdigão experiment was used for continuous RHI scans at a fixed angle. The results of these scans are presented in a different publication. Second, during the Perdigão experiment, the DLR WindScanners were suffering from a software bug, as described in the manuscript and could only be operated comparatively slowly, which limits the possibility to analyse turbulence to only the large scales. In the direct responses to the reviewer comments we will elaborate in detail on the mentioned issues and suggest to add some additional analysis to the revised manuscript, always highlighting the constraints of the dataset. Despite the constraints and limitations of the dataset, we still believe that the technique of measurements, the proof-of-concept and the results of the mean wind speed deficit is of high value for the scientific community and suitable for publication in AMT. It will hopefully foster future long-term campaigns with similar scanning strategies and smart layouts of experiments.

**1.1  RC2, General Comments**

- *I do not fully understand why two lidars have been sited close to each other and the third one further away. I would expect the greatest flexibility in measuring fluctuating wakes when having the lidars in a sort of an equally-sided triangle around the expected measurement volume.*
  As described in Sect. 2.2 of the manuscript, a primary scanning strategy that

was pursued for the experiment was RHI scans with all three systems. The advantage of these measurements over the experimental strategy described in the manuscript is that these are well-established methods with little risk of problems with software or hardware of the lidars. Another advantage is that these RHI scans were of great benefit for other research goals of the experiment, namely the characterisation of the flow in the valley and above. For wake characterization, unique two-dimensional flow visualizations of the wake could be captured for main wind direction as presented in Wildmann et al. (2018). For those coplanar measurements it was important to have one lidar (#2) in the valley measuring radial wind speeds at a higher elevation angle compared to the lidar on the North-East ridge (#1). The siting of the lidars is always subject to logistical constraints, especially in a complex terrain as in Perdigão (see Vasiljević et al., 2017). In best case, lidar #2 would have been placed closer to the wind turbine and at a lower elevation, but the topography and availability of electrical power did not allow it.
In a revised manuscript we will elaborate more on the siting of the lidars in Sect. 2.2.

- *I do not fully understand why the results have been discussed in terms of the Jensen Park model. This model has been developed for flat terrain and essentially neutral thermal stratification.*
We are aware that the Jensen-Park model has not been developed for complex situations as found in Perdigão and have mentioned it in the text. We still believe that there is a value of showing the prediction of this model in comparison to the measurement results to give the reader some reference. We are not aware of any engineering models that could take complex terrain, flow and atmospheric conditions as they are found in Perdigão into account.

- *The formula given for kw essentially says that kw is equal to turbulence intensity. Thus, turbulence measurements could be used to test the validity of the calculation of kw.*

Assuming that friction velocity $u_*$ is proportional to $\sigma_u$, it is true that $k_w \approx 0.4I$ (Peña et al., 2016). Again, theoretically this relation only holds for flat and homogeneous terrain such as we do not have in Perdigão. Nevertheless we can derive the theoretical wake decay for the wind turbine in flat terrain as a comparison to our measurements. As can be seen from Fig. 1, background turbulence intensity is 16% for 17 May and 13% for 2 June, which translates to a value 0.062 and 0.052 for $k_w$ respectively. This is considerably smaller than using the parametrization of the logarithmic wind profile ($k_w = 0.11$) and thus also lifts the model curve to almost the same level as the measurements (see Figs. 3-4). Section 4.3.1 is adapted accordingly in the revised manuscript.

- *For the assessment of the samples shown, it would be really interesting to learn something about thermal stability and overall turbulence intensity during these measurement periods. Maybe, this would be the clue to the overestimation or underestimation of the wake.*
  For a proper study of wake behaviour in different stability conditions, not enough measurements have been done with the described method, so that no statistics could be derived. A first attempt to classify the wake behaviour in Perdigão for different stability is described in Menke et al. (2018) and Wildmann et al. (2018). We will add some analysis on turbulence intensity from the lidar scans in the revised manuscript as Sect. 4.3.2. Figures 1-2 shows the result. On both days that are analysed, background turbulence intensity is of the same magnitude. On 17 May, it varies more, because measurements are carried out from late afternoon throughout the night until early morning. It is seen that in all cases $I$ in the wake is significantly higher and decays towards 8 D downstream where it reaches the background value.
**References**

Menke, R., Vasiljević, N., Hansen, K., Hahmann, A., and Mann, J.: Does the wind turbine wake follow the topography? - A multi-lidar study in complex terrain, Wind Energy Sci., submitted, 2018.

Peña, A., Réthoré, P.-E., and van der Laan, M. P.: On the application of the Jensen wake model using a turbulence-dependent wake decay coefficient: the Sexbierum case, Wind Energy, 19, 763–776, doi:10.1002/we.1863, we.1863, 2016.

Vasiljević, N., L. M. Palma, J. M., Angelou, N., Carlos Matos, J., Menke, R., Lea, G., Mann, J., Courtney, M., Frölen Ribeiro, L., and M. G. C. Gomes, V. M.: Perdigão 2015: methodology for atmospheric multi-Doppler lidar experiments, Atmospheric Measurement Techniques, 10, 3463–3483, doi:10.5194/amt-10-3463-2017, 2017.

Wildmann, N., Kigle, S., and Gerz, T.: Coplanar lidar measurement of a single wind energy converter wake in distinct atmospheric stability regimes at the Perdigão 2017 experiment., Journal of Physics: Conference Series, doi:submitted, 2018.

[Figure]

[Figure]

**Fig. 1.** Turbulence intensity in dependency of distance to the wind turbine for half hour periods (grey lines) and overall average (blue line) on 17 May.

**Fig. 2.** Turbulence intensity in dependency of distance to the wind turbine for half hour periods (grey lines) and overall average (blue line) on 2 June.

Interactive
comment

**Fig. 3.** Wind speed deficit in the wake of the wind turbine over distance for 17 May.

**Fig. 4.** Wind speed deficit in the wake of the wind turbine over distance for 2 June.